# Communal Organizational Culture as a Source of Business-Success Sustainability in Kibbutz Industry—Two Case Studies

**Yaffa Moskovich**

Zefat Academic College, Zefat 13206, Israel; yafam@zefat.ac.il

**Abstract:** This article explored the source of the organizational success of two kibbutz factories. This stood in stark contrast with most kibbutz industries, which abandoned their cooperative and communal attributes and became hierarchical and bureaucratic. This movement away from founding principles was the result of the massive privatization process that the kibbutz movement had been undergoing since the 1990s. This research followed the case study approach, with a comparative analysis of the two kibbutz firms. The author interviewed fifty respondents and supplemented the data with document analysis. The findings in the two factories reflected their ability to assimilate capitalist features into their organizational culture while preserving much of their kibbutz nature, expressed in various cultural features. These factories preferred kibbutz members over outsiders and maintained much of the original organizational democracy and equality among managers and workers. In addition, the firms demonstrated concern for the well-being of all the workers. This mixture of expertise and professionalism, along with internal democracy, equity, and communal concern, could be an example for other factories seeking business success. One important theoretical implication of this research is that an organization whose members identify with their organizational roots can better achieve long-term business success. Finally, this article offers practical implications for managers seeking to design a strong organizational culture.

**Keywords:** organizational culture; organizational success; business sustainability; kibbutz industry

## 1. Introduction

This article examines the impact of organizational culture on the sustainability of business success in the kibbutz industry, a perspective that is relatively new in the literature [1–6]. The central claim of the current research is that the communal origins and maintenance of key cooperative values of two particular factories facilitated their business success. This claim questions the prevailing assumptions that traditional kibbutz values cannot foster economic prosperity. Some kibbutz researchers have asserted that the process of privatization in kibbutzim (the plural of "kibbutz") and their factories is the inevitable result of the failure of socialist values in neo-liberal reality [7,8]. The current article balances the discussion by presenting cases where communal and cooperative cultural features serve as a source of organizational and economic success in the long run.

Young Jewish pioneers established the first successful kibbutz in Turkish Palestine in 1909. The kibbutz movement in Israel sprang from socialist principles [9,10]. The first kibbutz members, who came to Turkish (and then British) Palestine at the beginning of the 20th century, based their communities on equality and democracy. Many of them learned organizational skills during this period. The pioneers shared their property equally and strongly believed in the value of the sanctity of work [7]. They established democratic institutions, like the general assembly, where each decision in the community was the result of direct democracy [11]. The members incorporated these socialist traits into kibbutz factories, which first appeared in 1920 [12].

One of the basic values of the traditional kibbutz (i.e., before the wave of privatization) was socialism. Each member would contribute to the community according to his ability and would receive his needs from that community, irrespective of the market value of his work. The status of community members would be equal. They would share the work, and the communal assets would belong to all the members. This communal attitude applied to almost all aspects of life. In most of the kibbutzim established before the 1980s, children slept in age cohorts in special educational buildings and were with their parents in the late afternoon and evening. Other examples of communal life included the communal preparation of food and eating, education, and laundry [7–9]. Kibbutz members believed in gender equality, with men and women often sharing physical work and other communal duties, despite the tendency towards gender-specific types of work. Although the earliest kibbutzim were totally agricultural, in the 1920s, they began establishing an industry to provide work for members [10]. There was communal ownership of all agricultural and industrial means of production, along with egalitarianism, democracy, and rotating leadership [11].

Most kibbutzim identified politically with one of the various labor parties, which usually participated in, and dominated, the center-left coalition governments until 1977. In return, those governments supported the kibbutz movement, often covering the economic losses of particular kibbutzim. Those governments saw value in the kibbutz movement beyond economics alone: the kibbutzim (along with the cooperative villages called "moshavim") settled and developed rural areas; aided in immigrant absorption in various ways; and provided social, political, and military leadership far beyond the kibbutzim's [plural of one kibbutz] percentage in the general population. Nevertheless, two major events diminished kibbutz ties with the labor parties and the government, thus decreasing the importance of the kibbutz movement in Israel in general. First, the establishment of the State of Israel in 1948 saw the transfer of many functions from the kibbutzim (and other voluntary organizations) to state bureaucracies. Second, the formation of the first center-right coalition, after the 1977 electoral success of the Likud party, significantly weakened the ties between the socialist kibbutzim and the government. The new political realities exacerbated the economic crisis that the kibbutzim suffered during the following decade [7,9,10].

The economic crisis of the 1980s and 1990s resulted in a fundamental re-assessment of the underlying socialist values and practices in the kibbutz movement. The outcome was a massive process of privatization in the national kibbutz institutions, in most of the kibbutz communities, and in their respective factories [8]. Most kibbutz industries adopted more business-orientated practices, among which was the abandoning of equality and democracy in the workplace. The organizational structure of these kibbutz factories became hierarchic and autocratic [7,8]. Although these radical transformations saved some of the firms from financial collapse, at the same time, they caused human-resource problems [8]. In short, after the privatization of kibbutzim, their factories became similar to other enterprises in Israel [9–11].

For many decades, the kibbutz industry zealously kept to socialist principles; revenue and profit were not the primary goals of the factories [12]. Kibbutzim established and maintained factories to provide their members with employment as the population outgrew the communities' agricultural base. In most kibbutz factories, the members opposed hiring outside workers, fearing the exploitation of people that were not among the collective owners of the means of production [8]. In addition, the general assembly filled managerial positions based on social connection and seniority in the community. These non-capitalist practices often led to nominating unqualified administrators to manage the factories, which caused losses in revenue [13,14]. Often other non-capitalist values shaped the behavior of the kibbutz industry. For instance, some kibbutz factories rejected opportunities to manufacture goods destined for Germany because of the Holocaust.

Today, the kibbutz movement includes 270 communities, of which approximately 90% have undergone the process of privatization [2]. With privatization, the kibbutzim separated their economic branches from direct community control. Independent boards of

directors, many of whom were outside experts, decided overall factory policies and left day-to-day decisions to the CEO, who often was not a member of the kibbutz. Capitalist profit maximization became the norm as socialist values quickly faded into history [8–10].

The research questions of this study are:

a. What are the communal attributes that contribute to the success of these factories?
b. What is the explanation for this phenomenon?

The article draws from the literature about communal culture and its effect on the sustainability of successful businesses. The paper presents the qualitative methods used to examine the two research questions. The findings analyze the organizational culture of the two factories and the sustainability of their successful operations. The discussion explains the findings concerning the theories in the literature. Finally, the conclusion responds to the research questions and presents practical and theoretical implications emerging from the research.

## 2. Literature Review

### 2.1. Introduction

The literature review focuses on several inter-connected theoretical concepts: organizational culture, the sustainability of organizational culture, and their impact on the sustainability of high performance, productivity, and successful business [3–6]. The organizational culture of the traditional (i.e., non-privatized) kibbutz factory is communal culture, where the kibbutz and all its members own the factory; where there is social responsibility towards the members, both those who are working in the factory and those who are not; and where there is a fairly democratic managerial system [9–11]. Sustainability of a factory's organizational culture springs from the internal ethos that strengthens the solidarity, motivation, and productivity of managers and workers [15–17]. Moreover, the literature review and the current research illuminate the contribution of the sustainability of communal culture on economic behavior and long-term business success.

According to neo-liberal expectations, organizational success reflects the development of the business, the constant growth of profits, and the expansion of the business in local and international markets. The good reputation of a business stems from its continuity and survival over a long period. All of these elements define the sustainability of its business success [4–6].

### 2.2. Communal Organizational Culture

Organizational culture is the common glue that holds members together [15,16] and has a huge influence on the behavior of workers and managers [17], as organizational culture has an impact on production. Managers can shape organizational culture, and by doing so, they set the reality of everyday life [18–20]. Schein's classic work [18] developed the iceberg model, which consists of overt and covert dimensions of organizational culture. These obvious and concealed dimensions are strongly connected in communal culture [19–21].

Communal organizations, like kibbutzim, are a subset of cooperatives [22,23] and alternative organizations [24,25]. Cooperatives operate with democratic procedures, with the members of the organization involved in the process of decision making [2]. There are democratic regulations for elections, which define the duration of the leader's term. In addition, the leader of the cooperative usually holds office for a relatively short period, after which another member leads the organization [2,25,26]. This process of rotation encourages leaders to respect and support the members, carrying out the democratically expressed will of those members. The structure of the cooperative is flatter than capitalist organizations, with a small and soft hierarchy. Another dominant trait in cooperative structures is equality between the leader and the members [7]. For instance, in kibbutzim, there are several forums and committees where kibbutz members can express their views. If they are members of that particular committee, they can vote on the issue. If dissatisfied with a decision in one forum, a kibbutz member can appeal to a higher forum, all the way to the general assembly. The communication about deliberations and actions should

be transparent and open to the community. These structures and procedures reflect the cooperative values of trust and mutual beliefs, which the members share [27,28].

### 2.3. Cultural Sustainability

Organizations can strive to achieve long-term sustainability in several ways. One of the most important is creating and maintaining strong environmental relationships. Management seeks good working relationships in internal settings, as well as nurturing positive relations with external organizations, including governmental authorities [29,30]. In turn, social and normative adaptation to these outside bodies and other external trends can strongly influence the culture of the organization [8,11].

A constructive managerial attitude towards the workers is crucial in enhancing cultural sustainability in the organization. This contradicts the neo-liberal definition of success, which focuses only on economic profit, usually at the expense of the workers. To boost profits, capitalist managers strive to reduce expenditures, which leads to hiring cheap labor [2].

Managers that know how to create a positive atmosphere, focusing on positive human-resource abilities, can foster a high level of motivation and productivity [16,24]. Obviously, the major way managerial attitude manifests itself is in working conditions. If workers receive fair salaries, with appropriate benefits, the effect of a successful managerial attitude will appear in many ways. One of them is the social solidarity among workers [2]. Positive human relationships increase identification with the business and decrease absenteeism in the workplace. There is a greater feeling of trust and positivity among the entire staff. This climate fosters innovation, the feeling of belonging, and a sense of family in the enterprise. Thus, leaders have a great impact on subordinates, particularly when they use a positive transformational style that nurtures workers and establishes close and informal ties [31,32]. Workers see such managers as mentors, who guide them and provide a friendly working environment [31–33].

On the other hand, when management does not know how to establish a good working relationship with labor, it will damage sustainability by creating mistrust and a climate of suspicion. This is particularly true when workers see their employers as exploitative.

In either case, the managerial style will shape organizational culture by constructing a reality that contains overt and covert layers. Some of the elements of the overt layer are symbols, patterns of behavior, and physical artifacts. The covert, underlying layer consists of norms, values, and basic assumptions [18,19]. An organization in which management fosters overt and covert layers that are supportive and transformational leadership [31] has a better chance of achieving cultural sustainability than more autocratic organizations.

### 2.4. Business Success and Sustainability

Several aspects define organizational success: organizational efficiency, effectiveness, institutionalism, and survival [32,33]. Absolute efficiency is the ability of an organization to use minimum input to produce maximum output. This definition examines only the economic cost. On the other hand, effectiveness reflects the ability of the organization to achieve its goals. Institutionalism considers the symbolic effect of the organization: its reputation in society and the organization's acceptance of that reputation. Finally, survival is the result of the organization's ability to exist for a long period, coping with internal and external challenges [32,33].

There are several coping mechanisms for businesses to face these challenges. One is improving competitiveness in the global market [28–30]. Another is to adopt policies and strategies that anticipate future developments and environmental trends [30]. Recently, this has meant adopting the green commodities strategy [34], establishing a more friendly relationship with the climate, and reducing carbon emissions and other pollutants. This strategy, and other strategies responsive to outside factors, can help the business to create a good reputation, which enhances economic survival [5,6,34].

Moreover, organizations can achieve survival by developing organizational agility to respond quickly to environmental demands [35–37] and by developing organizational learning, which fosters such agility [36,37]. This agility depends on a constant flow of communication from internal and external sources. A successful organizational culture enhances this agility by quickly responding to transformations in local and global markets. Such a culture demands that organizational members internalize characteristics of flexibility and creativity to recognize and accommodate new trends.

Open channels of communication among workers and managers are necessary to achieve organizational sustainability. Communication is also needed with outside actors like customers, governmental authorities, competing businesses, and other enterprises. The flow of communication influences the process of decision making, facilitating managers in understanding more precisely what changes are occurring and how to respond to them with adequate business strategies [38]. Managers can cope by establishing organizational contingency plans, ensuring their enterprises can accommodate economic, legal, and technological innovation, as well as cultural features of their surroundings [38].

## 3. Materials and Methods

This research used qualitative methods, analyzing two case studies. The multiple-case-studies method facilitated the comparison of cases that had similar features [39,40]. In-depth interviews and document analysis enabled the researcher to examine organizational events and phenomena that rose from the field and fit into categories, according to topic-thematic narratives [41]. These procedures helped the researcher discover the overt and covert cultural layers in the organizational settings [42]. Finally, these methods facilitated the understanding of the interaction between cultural features and business success in the two factories [39–42].

The researcher chose Factories A and B, the subjects of the case studies, because of the striking resemblance of their history. The factories produce low-tech items used in agriculture, like water supply equipment. Both factories began as small workshops to supply work for kibbutz members; their professionalism and innovation caused them to develop and expand, becoming global enterprises. The major difference is that Factory A is an industry in a communal kibbutz, while Factory B is in a privatized kibbutz.

Kibbutz A and Factory A: Eighty-five radical-socialist pioneers from Galicia established Kibbutz A in 1922. The community existed from agriculture and grew in membership. During the War of Independence, in 1948, many of the members died, and the kibbutz itself suffered an attack. After the war, the kibbutz outgrew its agricultural base and turned to industry. Today, the community has approximately 1270 members, whose high standard of living stems primarily from Factory A and secondarily from agriculture and other enterprises. The kibbutz is multi-generational, with some families having four generations living in fairly close proximity. The originally homogeneous population of the community has become more diversified with the recruiting of new members from various ethnic groups as well as the various educational and occupational interests of the growing population.

Kibbutz A adopted a "dormancy rule", which means that veteran members would have substantially more privileges and wealth if the community was privatized in the future. This rule prevents privatization because the younger members and new members would receive much less from breaking up the communal structure than maintaining that structure [2].

During the 1950s, Kibbutz A established Factory A to provide work for its members. At first, the factory produced plastic items for domestic use. With the adoption of injection technology, it began to manufacture helmets for the Israeli army and various agricultural items. After signing a contract with the John Deere tractor company, Factory A expanded its marketing worldwide. It produced a wide range of plastic items for industry, farming, and domestic use [2]. Today, Factory A is the third most successful kibbutz industry, with four plants in Israel. It employs 1400 workers, of whom 40% are kibbutz members. Outside of Israel, it has branches in Europe, America, Asia, and Australia. The company's success

enables all members of Kibbutz A to live very comfortably, rejecting privatization and maintaining collective ideas.

Kibbutz B and Factory B: In 1936, a small group of 30 pioneers, refugees from Germany, established Kibbutz B. The population grew and is now multi-generational. Today, the population is about 860 people, with about half of them being members of the kibbutz. Despite recent privatization, the kibbutz still has a (commercial) community dining room and supplies (for a fee) early education, limited health care, and other municipal services. The high standard of living attracts new people to the community, making it more heterogeneous in age, education, profession, and ethnicity. As a "renewed kibbutz" (i.e., privatized kibbutz), it allows more individual freedom for the members. They can work inside or outside of the kibbutz but must pay communal taxes for municipal services and mutual aid [13,14].

Kibbutz B established Factory B in 1964 in partnership with another kibbutz. The factory produces oil-pipe valves. After adopting the high-quality European and American standards of ISO 9000 and 9001, it penetrated the global market. One of its well-known projects provided hundreds of construction levers for the tunnel linking England and France [1]. This factory became very successful by opening branches all over the world.

### 3.1. The Interviews

The current research included 56 interviews of respondents with various organizational positions. There were various representatives from each factory, encompassing top managerial positions, middle managerial positions, staff positions, and workers from the production line. The respondents were both kibbutz members and outsiders. The researcher collected the interview data using the "snowball technique", with each interviewee recommending potential interviewees relevant to the research topics. The interviews were generally open but did have a questionnaire to fall back on. After a pilot group, the researchers examined the original questionnaire and made modifications for subsequent interviews, with versions for each type of respondent (see Appendix A). Nevertheless, many of the interviews developed into free-form, friendly conversations, leaving the questionnaire aside. The researcher recorded the interview, if the respondent agreed, and summarized the interview after it was done. The interviewees seemed unrestrained, talked freely, and openly described the reality in the factories. The researcher asked about the work routine in the factory: the ways of communication, human relationships in the factory, and the atmosphere at work. These questions attempted to identify and define the industrial culture in each firm and determine the culture's effect on business success [37].

Using the multiple-case study method, researchers could compare sociological phenomena occurring at different periods [39,40]. Following this model, the majority of interviews occurred in Factory B during 2014–2015 and in Factory A during 2017–2018. The researcher had further discussions with some dominant figures in Factory B in 2017 and 2018. After completing the interviews, the process of analyzing and dividing the data into categories and themes took a considerable amount of time.

As a follow-up, in 2021, the researcher interviewed top figures in Factory B: the former president, the current finance manager, and the current production manager. They reported on the central development of Factory B at the time: Factory A had purchased 25% of the ownership rights. In these interviews, conducted and recorded via Zoom because of COVID-19, the leaders of Factory B explained the reasons for this move. In addition, in 2023, the researcher interviewed the CEO and the manager of the human resources department of Factory A to receive current information about the developments in that firm.

Factory A—Twenty-seven workers sat for interviews from 2017 to 2018. The top and middle managers all came from the kibbutz community. The senior managers consisted of the current CEO, a former CEO, and the current vice president. The middle managers were a former marketing manager, a current marketing manager, a purchasing manager, a finance manager, an Israeli site manager, a quality manager, a current human resources (HR) manager, a former HR manager, a laboratory manager, a purchasing manager, a

salary manager, a housewares-department manager, and a storage manager. Most of the staff members interviewed were from the kibbutz: a bookkeeper, two technicians, and a graphics-department employee. In addition, there were two engineers hired from outside the kibbutz: one from the quality department and the other from the development department. The production-line workers included: two temporary workers from the kibbutz and three workers hired from the outside.

Factory B—The respondents in Factory B sat for 29 interviews from 2014 to 2015. The senior respondents, all kibbutz members, were the current CEO, two former CEOs, the president, the vice president, and two directors. Middle-level managers participating in the research were also all kibbutz members, coming from the following departments: production, assembly, marketing, finance, logistics, painting, and human resources. Respondents with staff positions included both kibbutz members and hired workers from outside the kibbutz. The kibbutz members included two engineers, two quality department workers, and two bookkeepers. The hired staff workers were from the following departments: development, information systems, sales, and computers. In addition, five production-line workers, all of whom were kibbutz members, sat for interviews.

Table 1 presents a comparison of the respondents from the two factories, focusing on the number of kibbutz members versus the outside workers at each level of the organizational hierarchy. It is important to note that given the snowball technique used to choose most of the potential interviewees, the fact that the overwhelming majority of workers come from the kibbutz probably reflects the reality of the workforce in both factories. In addition, it is notable that all senior and middle managers were kibbutz members.

**Table 1.** Kibbutz Members and Outsiders According to Organizational Hierarchy.

| Level in Organizational Hierarchy | Factory A | | Factory B | |
|---|---|---|---|---|
| | Kibbutz Members | Outside Workers | Kibbutz Members | Outside Workers |
| Senior Managers | 3 | | 7 | |
| Middle Managers | 13 | | 7 | |
| Staff Workers | 4 | 2 | 6 | 4 |
| Production-Line Workers | 2 | 3 | 5 | |
| Total | 22 | 5 | 25 | 4 |

### 3.2. Document Analysis

Document analysis supplemented the data collected during the interviews. Factory A provided two booklets. The first, from 2020, summarized the history of the factory during its first sixty years of existence, supplying various information about the founders, the ideas of senior managers, and their strategies [43,44]. The first booklet also presented a great deal of information about the factory's organizational culture as well as the growth and development of the company. Despite its managerial perspective, the document provided general history and outlined the important stages in the factory's organizational life. In addition, an earlier booklet, from 2016 [44], completed the picture and provided the changes in the workforce during 2014–2015, business strategies, a record of in-house celebrations and local events, as well as the recording of opinions of various managers and workers. In short, it chronicled the common beliefs and practices during many years of prosperity. Both these booklets helped the researcher analyze the factory's organizational culture and to understand its impact on the factory's business success.

During the years 2005–2014, Factory B supplied many more documents [45,46] than Factory A did during its period of fieldwork. Booklets and pamphlets from Factory B provided a detailed picture of the significant events. An important booklet titled "The Dream and Practice" [47] summarized the first forty years of the firm. This booklet, and other documents, presented interviews with senior managers, which facilitated the analysis

of the cultural views of these leaders. The documents also chronicled organizational events and celebrations in the factory. The pamphlets supplied information about the economic development and the growth of the global business of the factory. As in the case of Factory A, the document analysis at Factory B supplemented and enriched the organizational data for the researcher [2].

To avoid bias, the researcher used a triangulation process to examine the reliability of the documents that came from the two firms [43–47]. It was obvious that the booklets presented the managerial perspective and offered a subjective picture of excellent organizations without flaws and criticism. To overcome this obstacle, the researcher interviewed all types of informants, from all ranks within the factories' hierarchies, non-workers from the adjacent kibbutz community, and workers who came from outside of the kibbutz. The data emerging from the interviews supported the information from the booklets and confirmed a climate that tended towards egalitarianism in the factories. Moreover, the researcher collected daily general and economic newspapers to update the information about the two firms. These external newspapers confirmed the booklets' descriptions of the businesses and the major developments reinforcing their organizational sustainability [48–54].

*3.3. Data Analysis*

The ground-field theory was the basis of data collection in this study [39–42]. The researcher began the research by interviewing organizational members, without having any organizational theory (beyond the obvious fact that the two factories were somewhat similar). The process began with the dynamic gathering of all pieces of information. After analyzing the organizational data, the researcher decided on the theoretical cultural direction that facilitated organizational success in the two kibbutz industries. The researcher also adopted the constant comparison method because of the great similarity in the cultural features of the two factories. The main difference was that one was in a communal community, while the other existed in a privatized community [14]. Nevertheless, the different organizational settings created only a minor cultural distinction. The researcher found common cultural features in both factories that led to economic success [3–5].

The comparative analysis of the data developed in the following stages:

1.  In the first stage, the researcher chose two kibbutz factories that engaged in low-tech production for agriculture and then proceeded to collect data.
2.  The second stage consisted of building categories with analytic themes. This occurred after the completion of the interviews. Some of these categories related to norms and cultural values, while other categories were themes about organizational success. The researcher sought patterns of behavior and habitual information in the organizational reality. According to Shkedi, this process offered "an accessible and theoretically flexible approach to analyzing qualitative data, locating it in relation to other qualitative analytic methods that search for themes or patterns, and in relation to different epistemological and ontological positions" [40] (p. 77).
    This second-stage process found meaningful overt and covert organizational culture, such as particular ethical codes of managers and workers, which the literature has recognized as "kibbutz DNA". This organizational culture includes equity, open communication, constructive criticism, tolerance, communal responsibility, a sense of family, and supportive leadership. The sustainability of organizational success emerged from several categories: organizational learning, innovation and professionalism, going global, and utilizing long-term strategies [41,42].
3.  The third, and last, stage was the matching of data to cultural theories [18–21] and concepts of the sustainability of organizational success [37,38].

## 4. Results

The findings of this research highlight kibbutz cultural features, which are cooperative characteristics that still dominate in both factories. These features include social equality despite hierarchical rank, friendly relationships, open and democratic communication,

social concern for the well-being of the workers, strong connections with the adjacent communities, stability of the factories, and successful dealing with external environmental factors.

### 4.1. Process of Decision Making

Today, the two factories have different decision-making processes, reflecting how they developed from their similar communal origins. Although a traditional (i.e., non-privatized) kibbutz owns Factory A, most decisions about the business no longer arrive at the kibbutz general assembly for discussion and approval. The management enjoys a high level of autonomy, with very little community intervention. One technician, who sat for an interview, was critical of the management's decreasing sensitivity towards public opinion in the adjacent kibbutz (whose members are the collective owners of the firm). The technician lamented the decline of democracy in factory decisions and opined that its roots were in the factory's phenomenal economic success. The resulting high standard of living lulled the kibbutz members into silent acceptance of the change. On the other hand, there were frequent instances of democratic decision making in middle management, where a manager would consult with the workers before making a decision. Interviewees from several departments echoed this point [2].

Factory B has developed a different decision-making process since the privatization of Kibbutz B. The policy decisions are separate from kibbutz committees and the general assembly. A board of directors controls the factory, and the CEO must abide by its directives. Nevertheless, top management needs to provide annual reports to the community. Like in Factory A, decisions in lower levels of Factory B are slightly more democratic, with middle-level managers often consulting with workers. However, in both factories managers have more authority than subordinate workers. There is an unambiguous hierarchy in each factory [1,2].

### 4.2. Social Equality despite Hierarchical Rank

Equal social relationships between workers and management, despite the obvious difference in organizational rank, were one of the important values in each factory. This was one of the basic cultural features in the narratives stemming from the socialist past of each kibbutz and its factory [1,2].

The hierarchy of the two plants was flat and soft. A human resources manager pointed to an atmosphere of relative freedom: a manager could not arbitrarily control the employees. Because the kibbutz members at all levels of the hierarchy met in the kibbutz dining hall, at local events, and at other kibbutz functions, they were accustomed to treating fellow members equally. This attitude extended to workers from outside the kibbutz as well. A production manager from Factory B noted that the managers hired from the outside usually could not adjust to this kind of relationship and quit.

The president of Factory B said that workers did not hesitate to come to his office and criticize his policies or decisions. He noted that during a conference one of his engineers stood up and spoke against a managerial policy. He stated that this was a feature of the democratic-socialist vision of the firm. In the same way, a production-line worker said that there was no distance between workers and management, as each worker felt free to share views and even criticize the management.

### 4.3. Friendly Relationships

Friendly relationships in the two factories stemmed from the sense of togetherness and belonging the kibbutz members shared in their communities, which owned the factories. Research has noted that this familial cultural feature was common in various kibbutz industries [17,24]. They had lived together for a long period and had maintained common values and beliefs. Reflecting the community values, a lab manager pointed out, "Our work is based on mutual assistance; each member feels a high commitment. Our department must give quick service if there is a fault in the production line: each worker that is

available will respond immediately". She added that when they were planning the shifts at work, they took into consideration if a worker had urgent issues outside the factory. This strong feeling of solidarity was acknowledged by a commercial manager, who said that the workers had a lot of enthusiasm for their work.

The sense of solidarity in the factories extended beyond friendship to the realm of familial relationships. Interviewees often presented this narrative in both factories because family members worked in the plants. In Factory B, the president's daughter worked in the position of human resource manager. During his interview, the president said that after his daughter had finished her master's degree in human resource management, she had many opportunities but chose to work in the kibbutz plant. Another example of family relationships in the factories was a married couple who worked as the finance manager and the production-line manager. In Factory A, the CEO and the human resource manager were related. The CEO further stated that his sister, brother, and wife had worked in the firm.

The familial climate in the factories sprang not only from the kinship relationships themselves but also from the feelings and acts of belonging. Kibbutz members brought cookies from home for fellow workers. When the office was dirty, the secretary cleaned the office, not waiting for the cleaning workers.

Both firms published newsletters, which informed the entire kibbutz community about current events in the factories, congratulate workers when they received promotions in the plant or had an important personal event, and welcomed new workers to the factory. The factories arranged holiday celebrations, family days, solidarity days, and other events in the factory and organized trips for all the workers to build the *esprit de corps* away from the factory itself. The newsletters presented articles and photos of these events to re-enforce the special familiar climate. In addition, management offered personal treatment for workers on their birthdays or when they suffered death in their families [2].

This feature of factory-as-family was also one of the reasons for the preference of kibbutz workers over outsiders. The norm of preferring kibbutz members was very strong in both factories, which first looked to the community for new workers for every position in the hierarchy. Only when the factory could not find the right person for a position did the managers begin to look for workers from the outside. As a result, most of the top managers and middle managers were from the local communities. In the interviews, the CEOs stated that they preferred kibbutz members because they had the kibbutz DNA, which an outsider did not always understand. The difference between locals and outsiders could cause all sorts of communication problems.

Both factories nurtured kibbutz members in low positions to rise to higher ones in the organization. For example, a former CEO had worked for 17 years in various positions before he became CEO. A former president opined that most outsiders did not suit the kibbutz DNA, which was why top managers came from the lower ranks. He said: "When you recruit from outside, there is always a risk that this manager is not suited to his job".

Thus, kibbutz workers felt that the factory belonged to them, but this perception could have complicated consequences. Some workers, especially those in professional positions, criticized this policy. They opined that kibbutz members felt too secure and assumed that no one would fire them: they did not have to work as hard as someone fearing termination. One staff member said, "Sometimes it is better to bring an outsider to the work. He is more professional and if you are not satisfied, you can fire him". Indeed, there were cases when the familial bond prevented managers from firing unqualified kibbutz workers.

On the one hand, kibbutz members (particularly relatives) were usually more committed to the enterprise and tended to work hard, but some kibbutz members were unqualified for their positions. In these cases, it was difficult for managers to fire them because the kibbutz general secretary (i.e., the leader of the kibbutz community) would usually support the member. In Factory B, for example, management transferred an unqualified worker from one department to another one but did not fire him. Similar events occurred in Factory A, for similar reasons.

## 4.4. Open and Democratic Communication

Transparent and democratic communication were norms in both factories [1,2]. The management in both factories embraced an "open door policy". Any worker could meet a manager and talk about what was happening and the new projects, without fear of retaliation. Despite busy schedules, CEOs would find time to discuss personal issues and work conditions. In the same vein, the president of Factory B said that workers could visit him and express their views. As the former president of Factory B said, "You can't connect only with WhatsApp: we are a place of people, we need human connections". He added, "We develop interpersonal communication and it is important to keep our good human resources".

The CEOs make a point to speak to workers when they visit the various departments. The human resource manager from Factory A noted that dissatisfied workers would not hesitate to express their feelings and "usually I will coordinate a meeting between the worker and his manager".

In both factories, there was "the forum" where the workers' representatives met with the human resources manager and discussed workers' problems and concerns. At these meetings, the management could also raise issues like low production and other worker-related problems. This meeting took place once a month and represented an indirect form of communication between management and workers.

Thus, direct and indirect lines of communication were open, transparent, democratic, and two-way. As a bookkeeper noted, there were various means of consultation: "We have mechanisms of checks and balances. Kibbutz members feel secure in their work; they can express their thoughts and views freely without threat of losing their job". The manager of the houseware department in Factory A added that she periodically shared information with her workers. Referring to when workers complained about their low salaries, she said, "I explained the reasons for the low pay". This policy of updates and explanations was common in other departments in Factory A.

Open and democratic communication was a key element of the socialist tradition of teamwork [9,11,12], even though managers obviously have more authority in the hierarchy. The group works together, sharing ideas. A human resources manager pointed out that in the teamwork meetings, they examined work plans together and received new ideas on how to improve procedures. A marketing manager in Factory A added that teamwork is sacred and efficient; the group was diverse, and each worker could contribute. As one manager said, "You need to leave your ego aside to hear what other workers are saying. In work meetings, workers participate and share their views".

## 4.5. Social Concern for the Well-Being of the Workers

The factories' familial organizational culture extended to caring for the well-being of their workers in specific and the kibbutz members in general. For example, Factory A hired elderly kibbutz members and provided them with suitable part-time work. Most of them worked in the plastic kitchen garments department. This aligned with the traditional kibbutz ethos of work being a vital part of a member's life as an individual and as a member of the community. In traditional (i.e., non-privatized) kibbutzim, retirement was not mandatory, and elderly members could obtain increasingly easier and shorter work assignments after the legal retirement age. In his interview, one of the older workers stated that he was happy to come to work because it supplied him with a daily routine that was important to him. His manager, from the kitchen garments department, said that when an older worker did not show up to work, she would make a wellness telephone call like a social worker would.

In Factory B, social concern appeared in different ways during the interviews. In one case, three workers got a truck, after working hours, and helped a fellow worker move apartments when going through a divorce. Other workers donated items to help him to adjust to his new situation. In another case, some workers helped renovate a house, painted the place, and donated paint and brushes.

Both factories hired young people from the kibbutzim right before or after mandatory army service. The factories did this even though temporary, inexperienced workers were less economically productive than other workers in the labor market. The managers accepted the kibbutz-wide opinion that these young people should have an opportunity to earn money for their private needs. Beyond that, some of these people could return to the factory after army service, a trip abroad, and obtaining a higher education. Thus, the managers saw these youngsters as potential workers and were willing to invest in them.

### 4.6. Strong Connections with the Adjacent Communities

Both factories were highly connected with the kibbutz community that owned them. Both of them physically existed in the kibbutzim, symbolizing their importance in communal life. As one of the kibbutz members noted, every major event in the factory was an important issue in kibbutz conversation. Every member was highly interested in the production and the workforce. Changes were topics of discussion in the communal dining room, at local events, as well at the general assemblies. The CEO of Factory B stressed that "the plant identifies with our community and the linkage is very strong". And because so many families had members working in the factories, most members felt this linkage.

### 4.7. Stability of the Factories

Both the factories had long-term policies to keep the factory operating in the future [2]. In part, this cultural norm stems from the value of "the sanctity of work" [7–10]. This was in contrast to some kibbutzim's decision to sell their factories and share the profit among the kibbutz members. This was particularly important for members of Kibbutz A, which had decided not to privatize. The managers of Factory A have repeatedly stated that the factory would continue to exist into the future and provide employment to the kibbutz members. The manager of the finance department said that the factory had an obligation to maintain operations into the future: "That is the belief of our members and we will keep this policy for the benefit of the members".

Despite Kibbutz B selling 25% of the ownership of Factory B in 2020 to Factory A, the managers and the kibbutz community stressed the importance of maintaining a controlling share of the factory into the future. The goal of the sale was to create financing for new investment in the factory, not to relinquish responsibility over the business with a total sell-out. In fact, to maintain the kibbutz culture in the factory, Kibbutz B turned to another kibbutz rather than a private venture. This was contrary to many kibbutzim who have recently sold off their factories to private investors [2]. Kibbutz B members still retain 75% ownership. Moreover, the other 25% in the hands of another financially solid kibbutz factory promises a continuation of Factory B's kibbutz culture into the future. The CEO of factory B said, "Kibbutzim are selling their factories. What interests them is only money; we are different. We want to keep our factory for the long run". The basis of this view was the fact that Kibbutz B maintained the characteristics of a cooperative community, although they sold a quarter of their industry.

To ensure the long-run sustainability and success of the two factories, the management teams adopted several strategies. The key strategies included excellence, professionalism, and high-quality standards; organizational learning; and innovation.

### 4.7.1. Excellence, Professionalism, and High-Quality Standards

Both factories knew how to internalize norms of excellence and high-quality standards. In the process, they modified traditional socialist values to the realities of the global capitalist market [2]. To ensure the long-term stability of the factories, the management strived to achieve and maintain internationally recognized quality standards. For example, adopting ISO 9000 and ISO 9001 allowed them to enter European and American markets. Upholding these high-quality standards was so important that they trumped the norm of protecting kibbutz members in the factory. If any worker's production was sub-standard, the factory would move him to another department or even fire him. Moreover, managers

from both factories stressed their priority of recruiting professional workers with high credentials to maintain and further the factories' achievements.

To maintain a high level of excellence in the international market, the factories had to be punctual in delivering their goods as well as preserving quality standards. The CEO of Factory B stated, "The competition in global markets is difficult; to survive you must be the best in the field". Managers of Factory A expressed similar ideas: to succeed meant keeping a good reputation and constantly striving for and achieving excellence in the field. The manager of the houseware department declared, "Excellence is embedded in the factory's DNA".

### 4.7.2. Organizational Learning

If a customer complained about something, the factories would check the issue and explore how to improve their products and service. The factories strived to learn from their mistakes, aiming for excellence and constant improvement. But beyond reacting to customer complaints and past errors, both factories invested a great deal of time and effort in becoming proactive learning organizations: assimilating new technologies and profiting from the experience of other successful firms.

Both factories were aware of the importance of improving organizational skills by learning procedures. The factories' learning strategies were diversified: some were organized in-house at the Israeli sites, some in the field at foreign branches, and some at pre-existing programs at universities throughout the world. Both firms sent engineers to learn new techniques at the Technion, at other Israeli schools, and at academic institutions abroad. Moreover, the factories sent their representatives to their branches abroad to guide and teach them new technologies and also how to solve production problems. Both dispatched workers to conferences, conventions, exhibitions, and workshops that dealt with new products. Learning seminars took place in the United States, China, and Spain. In addition to the external learning, there were numerous internal courses in the factories, which a human resources manager pointed to as requirements for workers to maintain excellence [37].

### 4.7.3. Innovation

Innovation was a common theme in the interviews and documents at both factories. Factory B opened an innovation unit and hired additional engineers [45–47]. The factory developed municipal water systems that were less expensive than pre-existing ones. It also developed new valves with high-tech elements for various uses: agriculture, industry, the home, and fire-fighting systems. These were stronger and better valves than had existed in the market. The president of the factory said, "The concept of the factory is changing to become a professional global firm. This requires us to invent cheaper and stronger valves". In 2021 the company developed a new filter system for the American market, called "Spin Klin Nova", which improved the quality of the water.

Factory A also established a department dedicated to innovation, bringing in new professionals and high-quality engineers. Before launching a new product, this department tested it again to ensure high quality, even after the product had received approval from the quality-control department [2]. Among its numerous achievements, Factory A improved cotton envelope-packing and produced "Pallet Mesh Flex" hay nets according to the specifications of the John Deere Company. Factory A's new plastic wires were stronger and cheaper than those previously on the market. In addition, the factory improved plastic storage containers and eating utensils for airplanes as well as plastic items for domestic use. The marketing manager stated, "We are very flexible and adapt our products to the customers' wishes". The CEO stressed that innovation was a key element in maintaining Factory A's role as an important player in the global market [43,44].

*4.8. Successful Dealing with External Environmental Factors*

Both firms developed a culture of agility [35–37] to respond to the challenges of external environmental factors, particularly in global markets. The factories adopted various strategies: two of the most successful being developing partnerships and opening foreign branches.

Both factories developed partnership agreements with other entities, in one form or another. In 1970, Factory A created a partnership with another kibbutz for ideological reasons: to supply work to their respective kibbutz members. Factory A maintained 75% ownership, and the other kibbutz received 25%. Another successful partnership that Factory A entered into was with John Deere in 1988. This agreement defined Factory A as the exclusive supplier of hay nets for John Deere. All the interviewees mentioned the benefits stemming from this agreement and noted its significant contribution to the prosperity of the factory. Similarly, Kibbutz B signed an agreement in 1978 with another kibbutz to share the ownership of Factory B: Kibbutz B would own 75% and the other kibbutz 25%. An additional partnership appeared when Factory A bought 25% ownership of Factory B in 2020.

Each company intended to break into global markets early in its development. Factory B opened a branch in California in 1977 and a logistics center in Shanghai a logistic center the following year. In subsequent years the factory established production branches or marketing offices in Australia, Brazil, Chile, Colombia, Egypt, Italy, Mexico, Peru, the United Kingdom, and additional branches in China. As a result, potential customers could buy the products of Factory B in more than 86 countries.

Factory A began ramping up its global strategy during the 1990s, basing its expansion on the 1988 contract with John Deere. After its American branch, the factory established branches in other parts of the world: England in 1995, Italy in 1999, Germany in 2000, Ireland and Canada in 2007, and Poland and Hungary in 2009. By 2022, Factory A was operating production sites around the world, manufacturing garments, plastic rollers, nets for hay, and other items originally designed in the Israeli plant [43,44].

## 5. Discussion

This research found a strong connection between a cooperative/socialist ethos and the sustainability of business success in these two factories. Several factors could explain this finding [1,2].

First, democratic management promotes economic success because the existence of open communication is important for achieving professionalism and excellence [4–6]. Kibbutz members in the two factories have the confidence to express their views to their organizational superiors without fear of retribution. This norm does not exist in most other enterprises, where workers are afraid to share antagonistic views. Capitalist managers often view outspoken workers, even with constructive criticism, as trouble-makers and miss out on the workers' experience and expertise.

Open communication in a friendly and trust-based environment supports the conditions for creativity and innovation. This is common in both factories [14]. In these successful firms, managers allow and even encourage worker feedback and participation in the decision-making process [2]. To achieve this, the management preserves much of the traditional socialist ethos that echoes the cooperative lifestyle of the kibbutz members. This kind of leadership enhances organizational solidarity and the familial nature of the organizational culture [31,32], which in turn leads to the sustainability of organizational success. Although some research indicates that high solidarity can cause conformism and groupthink [55], the result in these two factories is different. In a cooperative environment, workers can openly think differently in a way that was beneficial for the factories' success [1,2].

Nevertheless, the process of decision making has become less democratic. The development of stricter hierarchies accompanies the decrease in democratic decision making in top management. Kibbutz members have not seriously challenged this tendency be-

cause successful enterprises provide a high standard of living for the community. On the other hand, there are still elements of democratic decision making in various departments. Because both factories embrace the norm of teamwork, a wide range of workers discuss various issues at the department level [12].

Second, the two factories' structuring themselves as learning organizations increases the ability to successfully confront challenges both in and outside of the organization. The management not only considers the diversity of opinions of its workers [1], but it also responds to customer requests and complaints. governmental regulation and changing market realities. Both factories have innovation units, encourage professional training of workers throughout the firm, both in and outside of the factory, and have a network of overseas representatives who can learn the realities in each locality. As learning organizations, they increase their innovation and creativity as they further develop the agility and know-how to respond quickly to a wide range of challenges [31].

Third, the strong ties between the factories and the adjacent communities are a key component in the firms' success and sustainability. The adjacent kibbutzim, and thus all the members, are the owners of the factories. Because the managers have maintained much of the traditional cooperative and familiar culture of their communal past, the environment inside the firms tends to echo the environment of the surrounding communities. This strengthens the solidarity of the member-workers at all levels within the factories and their solidarity with the communities. On their part, most members of the kibbutzim are keenly aware of what happens in the factories because of official communications (e.g., newsletters, reports from kibbutz economic committees, as well as discussions in the general assembly) and informal reports from friends and family members who work there. The prosperity of the factories now and in the future contributes to the prosperity of every member of the respective communities. In short, member-workers strive to contribute to the factories' economic success, because the factories belong to them as kibbutz members [2]. This strengthens the long-term organizational success and sustainability of those factories.

It is important to note that most organizational literature has not discussed the sustainability of economic success of kibbutz industries from the cooperative perspective, instead has focused on the internal and external organizational features of highly privatized kibbutz factories. True, most of the privatized kibbutz factories have taken on capitalist norms and structures; Factory B has successfully resisted many of the drastic changes in the organizational culture, maintaining both economic success and sustainability—despite the privatization of its kibbutz/owner. On the other hand, there is a positive feedback loop between the non-privatized Kibbutz A and its economically successful factory, overcoming any calls for privatization. The stunning international success of Factory A allows Kibbutz A to prosper as a traditional communal entity, reinforcing the traditional communal values of the factory's management and workforce.

The current research innovates the pre-existing literature about the kibbutz industry by presenting a different and opposite direction in recent trends in kibbutz industries. Most of the literature about privatized factories has discussed stronger hierarchies, increased inequality, and the loss of other communal/democratic features [8]. The hypothesis of most of the pre-existing literature is that the socialist ethos in pre-privatized kibbutzim caused economic losses in their factories [2,13,14] and did not allow them to recover from the economic crisis of the 1980s and 1990s. In response to this crisis, the members chose to privatize most of the kibbutzim [2,14]. As part of this process, the factories adapted themselves to the external capitalist environment by drastically changing their organizational culture. The current research offers an opposite narrative, pointing to two highly successful kibbutz enterprises that have maintained the traditional kibbutz ethos. The cooperative/democratic values and structures of these two factories are key factors in the sustainability of their economic success.

## 6. Conclusions

This research found a very strong connection between organizational culture and the sustainability of organizational success in these two kibbutz firms [2–4]. The combination of communal tradition with business orientation fusing strong human relationships with high-quality business practices enhanced the sustainability of organizational success [23,28–30]. The explanation of these findings lies in the familial culture of the kibbutz's origins, which causes high identification of the members with their original ethos [7–11]. In short, kibbutz members tend to be more committed to the community and its enterprises than their urban counterparts [7–10].

Figure 1 demonstrates the interaction among three factors: 1. communal internal culture, 2. democratic management, and 3. competitive culture emphasizing elements of the external environment such as professionalism, innovation, and creativity [3–6]. These three factors facilitated the sustainability of the business success of the two kibbutz firms.

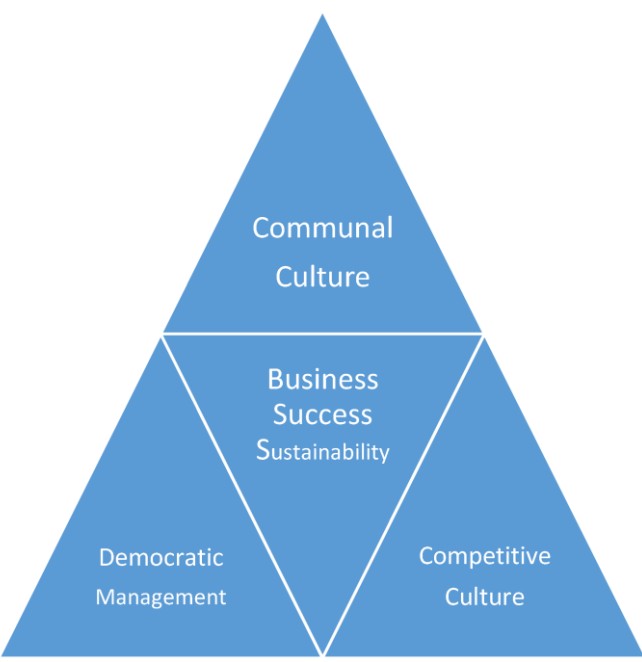

**Figure 1.** The Interaction among Communal Culture, Competitive Culture, Democratic Managerial Style, and Sustainable Business Success in the Two Kibbutz Firms.

This research innovates the existing literature about the kibbutz industry by stressing the cultural traits of the cooperative/democratic kibbutz ethos that fosters the economic success and organizational sustainability of two particular kibbutz factories [1,2]. Maintaining the cultural DNA of the traditional kibbutz is a defining factor in these factories' achievements. Other enterprises, whether in kibbutzim or not, whether privatized or not, can learn from these successful factories that one potential path to success and organizational sustainability includes a democratic culture, a relatively flat hierarchy, transparency of information, tolerance of criticism, a culture of organizational learning leading to innovation, and flexibility [2,7,8].

### 6.1. Theoretical Implications

1. Familial narratives and good working relationships can result in strong organizational culture.
2. A strong organizational culture can facilitate the sustainability of business success.
3. Strong ties between any business and its adjacent community are a key component in the firm's success and sustainability

4. Original communal cultural features of kibbutz enterprises included social responsibility, democratic management, equal status between managers and workers, open communication, stability in the long run, familial and friendly relationships, social concern for the well-being of the workers, and a strong connection with the adjacent community.

*6.2. Practical Implications*

1. It is important for management to nurture good human relationships.
2. A democratic managerial style can foster creativity and innovation in organizations.
3. Friendly relationships in organizations can improve production procedures.

*6.3. Research Limitations*

Like all qualitative research, there are limitations to arriving at strong generalizations from small studies [39]. In this case, there were only two cases providing data. Thus, it is impossible to make sweeping assumptions about other kibbutz enterprises, let alone non-kibbutz firms.

*6.4. Future Research*

It would be productive to conduct additional studies, asking the same types of research questions about other enterprises (kibbutz and non-kibbutz) to improve generalizations.

- To what extent do other kibbutz firms maintain their original democratic and communal ethos?
- To what extent does maintaining such a communal culture influence business success and sustainability in other kibbutz firms?
- To what extent does organizational culture influence non-kibbutz firms in Israel?
- To what extent does familiar organizational culture exist in other countries (e.g., Japan [55]) and what is its effect on business success and stability?

**Funding:** This research received no external funding.

**Acknowledgments:** Thanks to the Zefat Academic College for supporting this research.

**Conflicts of Interest:** The author declares no conflict of interest.

## Appendix A  Questionnaire

1. What is your organizational vision and how it was implemented in the factory? (managers).
2. Can you describe your policy about workers that are kibbutz members? (managers).
3. How do you recruit outside employees not from the kibbutz community? (managers).
4. Can you describe the organizational culture in the factory? (managers).
5. How is the general climate in the factory? (all interviewees).
6. Can you describe the channels of communication in the factory? (all interviewees).
7. How did socialist ethos shape the organizational culture? (all interviewees)
8. What is the kibbutz DNA and how did it influence organizational culture? (all interviewees)
9. What is the procedure for promotion in the factory? (all interviewees).
10. What are the criteria for managerial positions in the factory? (managers).
11. Are you satisfied with your working conditions? Please explain. (all interviewees).
12. Can you describe the relationship between managers and worker? Please give examples from your experience. (workers, not managers).
13. Can you give me an example of how communal responsibility influenced your work in the plant? (all interviewees)
14. What is the managerial attitude in the factory towards elderly workers? (workers, not managers).
15. What is the role of the firm in kibbutz life? (all interviewees).

16. Do you work with family members in the factory or with other kibbutz members? Can you give me an example? (workers)
17. How do you explain the success of your factory? (all interviewees).
18. Can you describe your business strategy? (only managers)
19. How did your firm become global? (all the interviewees).
20. As a worker, can you influence organizational decisions? Can you give me some examples? (workers)

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
