# Peer review of "Communal Organizational Culture as a Source of Business-Success Sustainability in Kibbutz Industry—Two Case Studies"

_sustainability, doi:10.3390/su151310654_

Round 1

Reviewer 1 Report

The topic is interesting, but the paper has some important flaws: 

First and most importantly, it has inconsistent conceptual and definitional frameworks. First, you start by declaring you are searching for the factors that enhance the economic success of the two factories. Later, you mention the cultural sustainability of an organization, without defining it. You only mention that it is opposed to the neo-liberal definition of success (lines 94-96) but you do not provide a definition per se. 

Then, you talk about „a successful organizational culture” (line 130) and „organizational sustainability”, without (again) defining them. Then, you mention a few times the „sustainability of organizational success” without defining what it is. 

This lack of clarity in conceptualization impedes the positioning of the article in a clear conceptual framework.  It just sends the message that the author is more focused on using some buzz words than drafting a clear conceptual framework in which to discuss the results of the research.

Secondly, the analysis seems to biased. Not only that the general tone of the paper is eulogistic (as opposed to objective and neutral) of every characteristic encountered in the organizational culture of the two case studies, but the documents mentioned at the Document analysis section are documents that are usually laudative of the management (in any organization), since they are commissioned by the management. This may be the cause why, when encountering situations that in other contexts would undobutably be construed as nepotism (hiring/promoting own family members) or discrimination (against those outside the community), in this article are presented in a bright light as friendly or family-like relations, or „preference of kibbutz workers over outsiders”. Of course that the persons who use unfair practices will defend them and explain why they are ok, but it is the responsibility of the researcher to make an objective assessment and pinpoint not just the strengths but also the weaknesses of the analyzed object.        

Thirdly, in my opinion, the results of the research are not sufficiently backed up by a conceptual framework specific to organizational culture or at least organizational management. 

There are also some minor improvements needed (after the major ones are implemented):

- the papers need a definition and a description and a listing of main characteristics of a kibbutz;

- there is no mentioning about what is the difference between kibbutz and kibbutzim, although it seems the two terms are distinct; 

- there is no presentation of the decision-making process in the two factories or the managerial roles; also, maybe a discussion of the managerial principles found in the two case studies on the backdrop of the general managerial principles would improve the paper;

- there is a lot of mentioning of the local communities where the workers and managers come from, but without describing the characteristics of these communities. What do they have „in common”/ makes them communities? Are they exclusionary to the „outsiders” based on nationality, religion, language spoken, geographical positioning? This could be something to be followed upon...

- the complicated ownership relations/transactions between the two case studies (lines 438, 509-510, 515-516) should be simplified as to be understandable;

- the idea of a „factory DNA”, repeated in many parts of the paper (sometimes in association with excellence), often in the vicinity of stories about preferential treatment for some workers that are in the inner circle vs. discretional firings of those that are not (“Sometimes it is better to bring an outsider to the work. He is more professional and if you are not satisfied, you can fire him.”) evokes strange vibes that I am not sure the author intends to; I recommend more wisdom in juxtaposing these ideas;

- not sure that references 57-58 and 60-63 belong there.    

The quality of the English language is good.

Reviewer 2 Report

Firstly, I would like to congratulate you on the development of this article. However, there are some improvements to be made.

Abstract. The author should clearly identify the main objective of this research and should state the implications for the field of this research.

Introduction. The research questions are clearly defined but the objective of the study should be too. The main purpose of the work should be stated. The structure of the paper should be described in the introduction.

Literature Review. The bibliography should be updated. For example, papers from 2021, 2022 are missing.

Methodology. The methodology of the interviews should be better described. A pilot test should have been carried out. There should have been an interview script describing the guiding questions. How were the responses registered? Were they recorded?

Something that is confusing is the date on which the interviews were carried out. The authors mention different dates for the two companies and compare them. This comparison is dubious. What is the current situation? What is the year to which this study refers?

Conclusions should be related with literature review, and it should be presented a comment to the objective. If the objective was achieved or not. The answers to the research question should be in conclusion. The conclusions should be supported by the results presented. The conclusions should be improved, and the results found should be commented on. The author should present theoretical and practical implications. Limitations and future research directions should be included.

It is missing appendix 1. The interview script as well as the questions asked should be placed in the appendix.

Reviewer 3 Report

First of all, thank you for the opportunity to review this interesting study which, from a purely qualitative approach, relates organisational culture and business success sustainability.

Second, I will point out aspects that, in my opinion, make it a relevant study that deserves to be published. The literature review is concise but sufficiently informative to frame the object of study. That is, it allows us to understand the differentiating characteristics of organisations such as the kibbutz, with a distinctly anti-coopeative/socialist ethos (in some ways historical singularities that deserve to be considered). Likewise, the Material and Methods section is clear, providing an overview of how the study proceeded.

Third, some issues that I think would have been relevant to consider in the paper. Regarding the method, I have missed concrete information on the analysis procedure (beyond the grouded theory approach). From the results and the discussion, I find the lack of nuances (the shadows), those aspects that always in a qualitative study balance between the positive and the negative (and make it differ from a panigyric). In other words, and by way of example, what findings have been found about possible limitations/disadvantages of the kibbutz culture to business sustainability or competitiveness? This ties in with an omission of 'Limitations' from the study that I think needs to be made explicit.

In summary, I consider this a relevant study that enriches the academic debate on the importance of organisational culture for business competitiveness.

Reviewer 4 Report

Title: Communal Organizational Culture as a Source of Business- Success Sustainability in Kibbutz Industry – Two Case Studies.

1.      At first, I think the authors should make clearer the connection of economic meaning with sustainable target.

2.      It would be useful if the authors explain better how green practices are incorporated in their analysis for deriving the sustainability. (See for instance Tsagkanos et al. 2022).

3.      It would also be useful for the audience and future researchers if a guide for the future research is provided: how this research could be used concretely to open new pathways? Is it possible to provide some examples and possible directions for future research? 

I think that a revised version with the abovementioned concerns could be a contribution to the literature.

Literature

Tsagkanos, Α., Sharma A., Ghosh B. (2022) “Green Bonds and Commodities: A new asymmetric sustainable relationship” Sustainability MDPI. 14, 6852.

The quality of English Language is fine. 

Round 2

Reviewer 1 Report

Congrats, you have improved your paper significantly and you argument is now more compelling. Good luck with your future research endeavors! 

Minor English language corrections are needed. Also, check the entire manuscript again, from start to finish.. Some paragraphs are repeated in different sections of the paper. 

Author Response

Dear reviewer,

  1. All the article was corrected to avoid redundancies! The article is now shorter than before. Before it had 11, 938 words and in the last version 11,625.

  1. The English language was re-edit by English native speaker - Doctor Jimmy Baker.

Best Wishes

Professor Yaffa Moskovich